# Incorporating Context into Language Encoding Models for fMRI

**Shailee Jain**[1]    **Alexander G Huth**[1,2]
Departments of [1]Computer Science & [2]Neuroscience
The University of Texas at Austin
Austin, TX 78751
{shailee, huth}@cs.utexas.edu

## Abstract

Language encoding models help explain language processing in the human brain by learning functions that predict brain responses from the language stimuli that elicited them. Current word embedding-based approaches treat each stimulus word independently and thus ignore the influence of context on language understanding. In this work, we instead build encoding models using rich contextual representations derived from an LSTM language model. Our models show a significant improvement in encoding performance relative to state-of-the-art embeddings in nearly every brain area. By varying the amount of context used in the models and providing the models with distorted context, we show that this improvement is due to a combination of better word embeddings learned by the LSTM language model and contextual information. We are also able to use our models to map context sensitivity across the cortex. These results suggest that LSTM language models learn high-level representations that are related to representations in the human brain.

## 1 Introduction

To extract meaning from natural speech, the human brain combines information about each word with previous words, or context. Without context, humans would be unable to understand homonyms, parse phrases, or resolve coreferences. Contextual information must thus be represented in the human cortex. One powerful tool for mapping cortical representations is encoding models, which use features extracted from stimuli to predict brain responses recorded with fMRI. Previous language encoding studies have successfully mapped word-level semantic representations using embedding vectors (15; 22; 9; 8), but this approach neglects the effect of context by assuming that the response to each word is independent. Overcoming this limitation would require extracting language features that incorporate context. However, no existing hand-designed feature space is well-suited for this task.

Several recent studies have overcome the limitations of hand-designed feature spaces by instead using representations discovered by supervised deep neural networks. These networks are trained to perform tasks that the human brain performs: recognize objects in visual scenes (2; 7; 5) or words and musical genres in sounds (11). Stimuli from fMRI experiments are fed into the pretrained networks and activations from each layer are treated as separate feature spaces for encoding models. These representations prove highly effective at modeling brain responses in the visual and auditory cortex, suggesting that the networks discover representations similar to those in the brain. However, in both cases, representations at both the lowest-level (images, sound spectrograms) and highest-level (image categories, words, music genres) are known *a priori*, and the networks are used to discover intermediate ones. In contrast, for language the highest-level representation is unknown, making this supervised approach inapplicable.

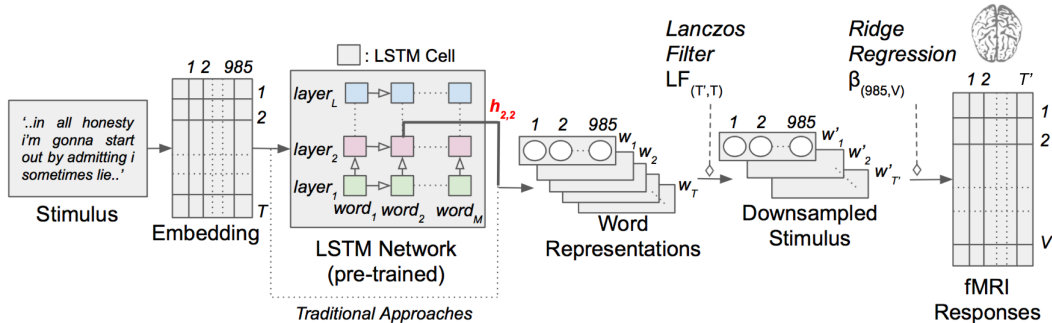

Figure 1: Contextual language encoding model with narrative stimuli. Each word in the story is first projected into a 985-dimensional embedding space. Sequences of word representations are then fed into an LSTM network that was pretrained as a language model. In this example, we extract representations for each word in the stimulus from layer 2 of the LSTM with context length 1 (i.e. considering only one word before the current word). A low-pass Lanczos filter is used to resample the contextual representations to the low temporal resolution of fMRI, and ridge regression is used to map downsampled contextual representations to fMRI responses. The dotted line indicates the path followed by the traditional, non-contextual approach that relies solely on individual word embeddings.

An alternative is to use self-supervised networks like neural language models (LMs). These models are trained to predict the next word in a sequence based on the previous words (4), typically using long short-term memory (LSTM) networks (21). LSTM LMs have been shown to discover high-level semantic and contextual representations that are useful for many natural language processing tasks (18; 13; 19). Further, a previous study suggests that representations learned by an LM can be useful for modeling MEG brain responses to reading (23). Thus, it seems plausible that LSTM LMs could discover contextual representations that would be effective for understanding language processing in the brain.

Another useful property of an LSTM LM is that it can generate many different types of representations that can be used to probe diverse representations in the brain. Firstly, different brain areas are known to have different 'temporal receptive fields' that indicate the amount of context they are sensitive to (12). This effect can be emulated in an LSTM by varying the number of words, or 'context length', used to generate representations. Secondly, LSTM LMs have multiple layers that can each capture distinct representations. Some previous natural language processing works use only the last layer (18; 13) while others use a linear combination across layers (19). However, previous fMRI studies have found that different layers from supervised networks predicted different brain areas (2; 7; 5; 11), suggesting that each layer should be tested separately.

In this work, we make two broad contributions. Firstly, we use representations discovered by an LSTM LM to incorporate context into encoding models that predict fMRI responses to natural, narrative speech (Figure 1). These contextual models perform significantly better at predicting brain responses than previously published word embedding models. Secondly, we compare the effectiveness of models that use different LSTM layers and context lengths. This reveals a hierarchy of brain areas that are sensitive to both different types of contextual information and different temporal receptive field sizes. Our LSTM-based contextual encoding models thus not only outperform the best hand-designed feature spaces for language encoding, but also provide insights into the representation of linguistic context in the cortex.

## 2 Natural Language fMRI Experiment

We use data from an fMRI experiment where the stimulus consisted of 11 naturally spoken narrative stories from *The Moth Radio Hour*, totalling over 2 hours or roughly 23,800 words. These data were used in previous publications and are described in detail there (9; 8). Each story is transcribed and the transcripts are aligned to the audio, revealing the exact time when each word is spoken. These rich and complex stimuli are highly representative of the language that humans perceive on a daily basis, and understanding these stimuli relies heavily on contextual information.

Measured brain responses consist of whole-brain blood-oxygen level dependent (BOLD) signals recorded from 6 subjects (2 female) using functional magnetic resonance imaging (fMRI), while they listened to stimuli. Images were obtained using gradient-echo EPI on a 3T Siemens TIM Trio scanner at the UC Berkeley Brain Imaging Center using a 32-channel volume coil, TR = 2.0045 seconds (yielding about 4,000 timepoints for each subject), TE = 31 ms, flip angle = 70 degrees, voxel size = $2.24 \times 2.24 \times 4.1$ mm (slice thickness = 3.5 mm with 18% slice gap), matrix size = $100 \times 100$, and 30 axial slices. All experiments were approved by the UC Berkeley Committee for the Protection of Human Subjects.

## 3  Voxelwise encoding models

Encoding models aim to approximate the function $f(S) \to R$ that maps a stimulus $S$ to observed brain responses $R$ (16; 24; 10). For natural language, $S$ is a continuous string of words $w_1, w_2, \ldots, w_T$. In fMRI experiments, a separate encoding model $\hat{f}_j$ is typically estimated for each voxel based on a training dataset, $\{S_{trn}, R_{trn}\}$. To evaluate model performance, the estimated model is used to predict responses in a separate testing dataset, $\hat{R}_{test,j} = \hat{f}_j(S_{test})$. Model performance for a single voxel is computed as the Pearson correlation coefficient between real and predicted responses, $r_j = \text{corr}(R_{test,j}, \hat{R}_{test,j})$.

In practice, the limited amount and quality of fMRI data cause generic nonlinear function approximators to perform poorly as encoding models. Instead, it is common to use *linearized* models, which assume that $f$ is a nonlinear transformation of the stimuli into some feature space followed by a linear projection, $f(S) := g(S)_{(T,P)}\beta_{(P,V)}$, where $g$ nonlinearly transforms $S$ into a $P$-dimensional feature space, and $\beta$ contains a separate set of $P$ linear weights for each of the $V$ voxels. The function $g$ is typically chosen to extract stimulus features that are thought to be represented in the brain, and the weights $\beta$ are learned using regularized linear regression. Here, we use ridge regression (L2-norm) to estimate weights for all encoding models, as described in a previous publication (9).

BOLD responses are low-pass relative to the stimuli that elicited them. To account for this effect we resample and low-pass filter the stimulus representation $g(S)_{(T,P)}$ to the same rate as the fMRI acquisition with the use of a Lanczos filter, yielding $g'(S)_{(T',P)}$. Then, to account for hemodynamic delay we use a finite impulse response (FIR) model with 4 delays (2, 4, 6, and 8 seconds), similar to previous work (9; 8).

### 3.1  Word embeddings for language encoding

The key question when constructing an encoding model is how to choose the nonlinear transformation $g$ that is used to represent the stimuli. The state-of-the-art language encoding model (9) uses a word embedding space that represents each word $w_i$ as a 985-dimensional vector $e_i$. These representations are based on word co-occurrence statistics from a large corpus of English text. (Although not identical, this embedding performs similarly to popular methods such as word2vec (14).) Word embedding spaces capture the semantic similarity between words, such that words with similar lexical meanings are represented by similar embedding vectors. Encoding models that use word embeddings have proven quite successful at predicting brain responses to both single words (15; 25) and continuous language (22; 9; 8; 17). However, one critical flaw is that these models assume that the brain response to each word is independent of the other words in the stimulus. Additionally, in most experiments the words are presented in isolation as opposed to a naturalistic narrative. These models treat the stimulus as a bag-of-words, ignoring temporal order and dependencies that are known to be significant for language processing in the brain (12). Our goal in this work is to find representations that bridge this gap by explicitly considering the order and dependency between words in a narrative.

## 4  Learning representations of context

Language models are trained to predict the next word in a sequence given information about previous words, i.e. for some sequence $[w_{i-m}, w_{i-m+1} \ldots w_i, w_{i+1}]$, they aim to maximize the probability $P(w_{i+1}|w_{i-m} \ldots w_i)$. To achieve this in an LSTM LM, representations of the words $[w_{i-m} \ldots w_i]$ are effectively combined in a hidden state $h_i$ that can be used to predict $w_{i+1}$. As a consequence,

$h_i$ must capture relevant contextual information (19) and might serve as a plausible candidate for modeling contextual representations in the brain.

LSTM LMs comprise LSTM cells arranged as a hierarchy of $L$ layers. For the word $w_i$, an LSTM cell at layer $l$ takes two inputs: a representation of the current word from the previous layer $(h_{i,l-1})$, and the state of the current LSTM cell after the previous word $(h_{i-1,l})$. The cell then performs a nonlinear transformation on these inputs to create a new word representation $(h_{i,l})$ that is both fed to the next layer and used to modify the state of the current cell for the next word. Thus, the hidden state for word $w_i$ at layer $l$ can be defined as

$$h_{i,l} = \begin{cases} LSTM(e_i, h_{i-1,1}) & \text{for } l = 1 \\ LSTM(h_{i,l-1}, h_{i-1,l}) & \text{otherwise,} \end{cases} \tag{1}$$

where $e_i$ is a pretrained embedding vector that is used as the input representation for $w_i$, and $LSTM$ denotes the nonlinear function applied by the LSTM cell on the current and previous words. Therefore, layer 1 makes direct use of the embedding representation while subsequent layers increasingly depend on recurrent states.

Before the LSTM can be used to extract contextual representations, it must be trained on the language modeling task. Our LSTM LM is pretrained on a corpus of comments scraped from reddit.com (9), comprising over 20 million words. We chose this particular corpus because the informal and conversational nature of the text is more similar to our stimulus stories than most conventional corpora. The vocabulary is restricted to include all the words in $S$ as well as the top 10,000 most frequent words from the corpus. This ensures that the network is aware of every word in $S$ and can assign some prediction probability to each. For the input representations $e_i$, we use the 985-dimensional word embedding from the state-of-the-art encoding model (9). Our selected LSTM LM architecture has $L = 3$ layers, each with 985-dimensional hidden states. This dimensionality was chosen to match that of the input embedding so that every encoding model would have the same number of parameters. The LM is trained on word sequences of length $M = 20$, as we observe little gain in perplexity with longer sequences. The model is implemented in TensorFlow (1) with cross-entropy loss. We use the RMSProp optimizer (20) with an initial learning rate of 0.0005 and inverse time decay at 0.7 decay rate. Additionally, we implement early-stopping when the loss starts to plateau and no dropout. After training the LSTM LM, we evaluate its performance using $S$ as a test set. Due to the limited amount of stories and their high diversity, we observe no benefit of fine-tuning the LM on a part of $S$ and evaluating on the rest.

### 4.1   Extracting contextual information

Our LSTM LM uses information from up to $M = 20$ previous words to construct $L = 3$ separate context representations. By varying the number of words the LSTM has seen before the representation is extracted, we can produce representations with different 'temporal receptive field' sizes (12). And by extracting representations from different layers we may be able to capture different types of high-level contextual representations (19).

We extract $M \times L = 60$ separate representations of the stimuli, one for each context length and at each layer, by forward-propagating the stimuli through the pretrained LSTM LM. To obtain context representations for a stimulus story $S = [w_1, \ldots, w_n]$ with a desired context length $m \in [0..M-1]$ and layer $l \in [1..L]$, we use the following procedure. First, for each word $w_i$ we extract the previous $m$ words to form a length $m + 1$ sequence $[w_{i-m}, \ldots, w_i]$. The $m$ context words and word $w_i$ are sequentially fed into the pretrained LSTM LM, and then the activations from layer $l$ of the LSTM LM are extracted. The resulting 985-dimensional vector combines the information in $w_i$ with the previous $m$ words through recurrent connections, building an effective contextual representation for $w_i$. This procedure thus allows us to incorporate and experiment with varying temporal fields and contextual information in language encoding.

## 5   Results

Here, we evaluate the performance of our contextual encoding models against the state-of-the-art embedding-based model and examine the effect of using different context layers and lengths. We test the importance of context by distorting it and observing the effect on encoding performance. Finally

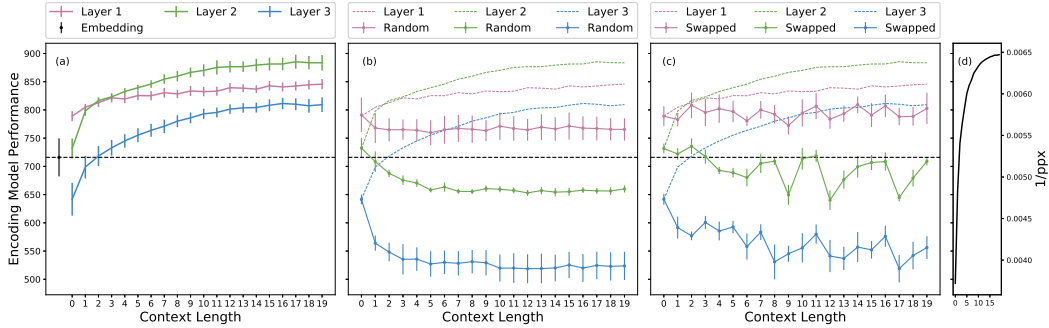

Figure 2: (**a**) Contextual encoding model performance with different context lengths and layers vs. state-of-the-art embedding. Results are averaged across 6 subjects ± adjusted standard error of the mean. Contextual models from all layers outperform the embedding. Increasing context length uniformly improves performance in every layer, and different performance across layers suggests that each represents different information. Best performance is obtained using layer 2 with long context. (**b**) To show that contextual models do indeed rely on context, we re-fit the encoding models with representations that used random words as context. Random-context models perform worse than the original models, and most perform worse than the embedding, except in layer 1. (**c**) Further, we swap context between different occurrences of each word and re-fit the encoding models. Swapped-context models also perform worse than the original, showing that linguistically plausible but wrong context also hurts the model. (**d**) Performance of the LSTM network on the language modeling task, as a function of context length. LM performance is measured by inverse perplexity (higher is better). Longer context leads to better LM performance.

we examine which brain areas are best fit by different context lengths and layers, revealing a temporal hierarchy in the human language processing system. All brain maps are made using pycortex (6).

## 5.1 Language encoding performance for context representations

After fitting the 60 contextual encoding models (with $M = 20$ different context lengths and $L = 3$ different layers) and a baseline embedding model from (9), we evaluate model performance by predicting fMRI responses on a test dataset comprising one 10-minute story. Model performance in subject $s$ is summarized by first computing $r_{sv}^2 = r_{sv} \times |r_{sv}|$, the fraction of variance explained by the model in the $v^{th}$ voxel. Then, we sum across all voxels in subject $s$, giving $r_s^2 = \Sigma_v(r_{sv}^2)$. Finally, to summarize model performance across the 6 subjects we average all the $r^2$ values, $r_{mean}^2 = \Sigma_s(r_s^2)/6$. Standard error of the mean (SEM) is used to compute error bars.

Figure 2(A) shows encoding model performance for each model, averaged across subjects. Contextual models significantly beat the embedding in nearly every model variation. This supports our hypothesis that contextual representations are a better match to the human brain than pure word embeddings. This effect is also seen in Figure 3, which shows the difference in performance between the contextual and embedding models for every voxel in one subject. While some voxels show little difference, most regions are considerably improved with the contextual model.

**Impact of context length.** Figure 2(A) shows that model performance increases monotonically with context length, for all layers. This suggests the encoding models are successfully exploiting contextual information that the LSTM LM has learned to extract. However, model performance plateaus after 10-15 words, suggesting that the LSTM LM was unable to successfully incorporate contextual information beyond this timescale.

This result is mirrored by the performance of the LSTM LM on the language modeling task, shown in Figure 2(C). Here, we test how well the LSTM LM predicts the next word from context on the stimulus $S$. For each context length $m$ we compute inverse perplexity, $PPX^{-1}(m) = e^{\Sigma_{i=1}^n \log( P(w_i|w_{i-m-1},\cdots,w_{i-1}) ) / n}$, where higher values mean better LM performance. The similarity between LM and encoding performance suggests that the ability of an LM to generate

Figure 3: Difference between contextual and embedding model performance across cortex. The difference in $r$ values is computed for each voxel in one subject (S1) and projected onto the subject's flattened cortical surface. White outlines show ROIs identified using other experiments. Dashed lines indicate cuts made to the cortical surface during the flattening procedure and the green stars indicate the apex of each cut. Red voxels are best modeled by the contextual model, blue are best modeled by the embedding, and white show no difference. Voxels poorly predicted by both models appear black. Overall, the contextual model performs better, although some voxels have approximately equal performance with both models.

useful encoding representations is tied to its language modeling ability. This suggests that better LMs yield better encoding models.

The encoding model results in Figure 2(A) also suggest that the improvement of our model can be partially explained by the LSTM LM simply learning a better word embedding than the baseline. Models with zero context length use no contextual information and thus are simply nonlinear transformations of the input embedding. The model for layer 1 with zero context is substantially better than the embedding, suggesting that layer 1 learns to generate a new embedding that is a better match to the brain. However, the best model for each layer uses the longest context, suggesting that contextual information is important above and beyond the improved embedding. We explore this issue further in Section 5.2.

**Impact of LSTM layers.** Figure 2(A) also shows that there are differences between models that use different LSTM layers. Further, the effect of context length is different for each layer. Layer 1 predicts substantially better than baseline even with short context, but only improves modestly with context length. Layer 2 predicts roughly the same as baseline with zero context, but jumps sharply with even one word of context and is the best model thereafter. Layer 3 predicts worse than the embedding with short context, but sees the largest improvement with context length. Investigating the differences between representations in different layers is an important direction for future work.

## 5.2 Distorting context representations

Figure 2(A) shows that the LSTM LM produces useful representations for encoding. However, it also shows that part of the improvement is due to the LM learning better embeddings, as can be seen by

the good performance of layer 1 with zero context. To directly test the importance of context in these models, we conduct two experiments where we degrade the quality of the context.

In the 'random' experiment, we modify the representation extraction procedure so that the context for word $w_i$ consists of a randomly chosen, nonsensical sequence of words rather than the words that actually precede $w_i$ in the stimulus. This yields a distorted representation for $w_i$, where the amount of distortion scales with the context length. Encoding model performance using these distorted representations is shown in Figure 2(B). It is obvious that the performance greatly suffers from the context distortion. Further, the impact on performance progressively increases across LSTM layers with layer 1 being affected the least, and still beating the baseline. Since the representations beyond layer 1 do not directly make use of the embedding and increasingly rely on contextual information passed through recurrence, the progressive impact across layers is justified. This demonstrates that the LSTM successfully incorporates context, although there are differences across layers.

In the 'swapping' experiment, for each stimulus word and context length we swap the actual context with the context of another occurrence of the same word (e.g. "this is my *dog*" → "I saw the *dog*"). Swapping is done after inferring context vectors using the LSTM but before regression on the fMRI data. For words that only appear once in the stimulus ($< 6\%$), we leave the original context in place. For each context length, we pick the new context that has the fewest words in common with the original. Thus, the swapped contexts picked for each successive length are sometimes different, leading to the increased variance across context lengths. The results in Figure 2(C) show that linguistically plausible but wrong contexts also greatly impact encoding performance. Additionally, we observe the same patterns of progressively greater impact across layers as in the random context models. This demonstrates even more effectively than before that the context model is picking up information relevant for cortical representations.

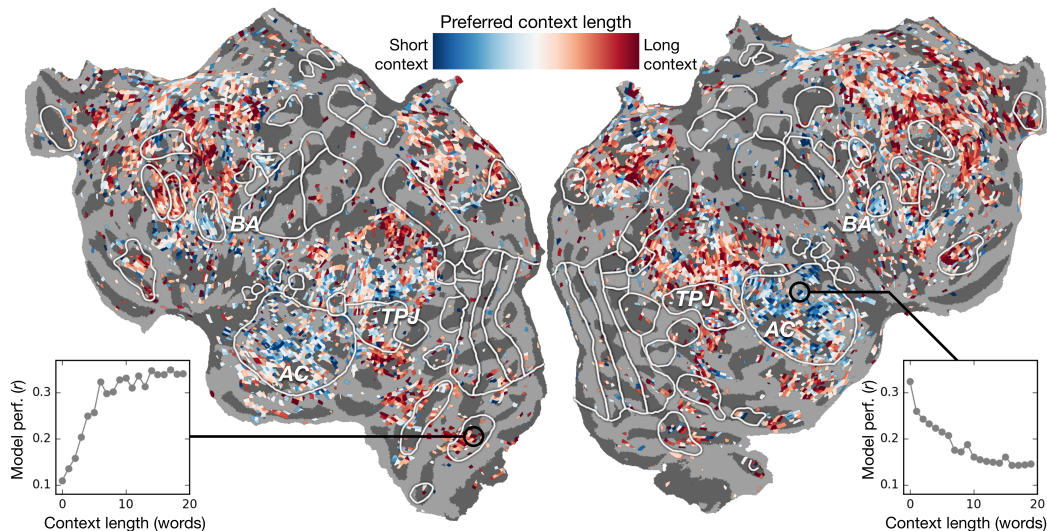

Figure 4: Context length preference across cortex. An index of context length preference is computed for each voxel in one subject (S1) and projected onto that subject's cortical surface. Voxels shown in blue are best modeled using short context, while red voxels are best modeled with long context. Non-significantly predicted voxels (mean $r < 0.11$) are gray. Insets show model performance with each context length for two representative voxels, one that prefers short context (right) and one long (left). Generally, voxels in low-level language areas (AC) prefer short context, while voxels in higher-level language areas prefer long.

## 5.3 Model preference across cortex

Finally, we use our models to investigate the importance of contextual information for representations in each area of the cortex.

**Context length preference across cortex.** Although longer context produces better encoding models overall, the same is not true for every brain area. To investigate this issue we compute a context length preference index separately for each voxel. This index is defined as the projection of each voxel's 'context profile', a length-$M$ vector containing prediction performance for each context length with LSTM layer 2, onto the first principal component of context profiles across all well-predicted voxels ($r > 0.15$). Layer 2 is used because it has the best performance overall; this analysis was repeated for the other layers and yielded similar results. This index was selected because it accounts for the fact that performance plateaus for context lengths between 10 and 20, and it avoids computing a noisy argmax across context lengths.

Figure 4 shows the context length preference for each voxel in one subject. Although most voxels prefer long context, clusters of short-context-preferring voxels appear in auditory cortex (AC), Broca's area, and left temporo-parietal junction (TPJ). These areas thus seem to selectively represent information about the current word rather than information that is built up with context. This supports the findings of (12), which showed that similar brain areas respond strongly even when a story is scrambled to remove context. In a previous study, Baldassano et al. (3) report findings that suggest different temporal scale selectivity across the cortex. Our contextual encoding model captures this timescale variance, but goes further by explicitly modeling the transformation from stimulus to response.

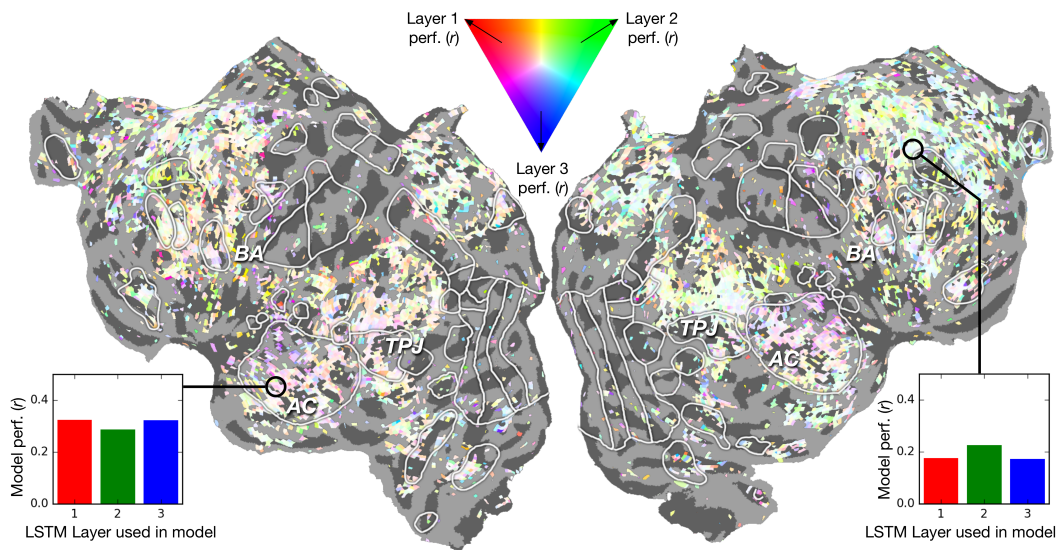

Figure 5: LSTM Layer preference across cortex. An index of layer preference is computed for each voxel in one subject (S1) and then projected onto that subject's cortical surface. Voxels that are much better-predicted by layer 1 than the others appear red; layer 2, green; and layer 3, blue. Voxels equally well-predicted by all layers appear white. Non-significantly predicted voxels (mean $r < 0.11$) are gray. Insets show model performance with each layer (averaged across context lengths) for two representative voxels, one that slightly prefers layers 1 & 3 (left), and one that slightly prefers layer 2 (right). For most voxels there is little difference in performance from different layers. However, there is a slight preference in AC for layers 1 & 3, and in higher semantic regions for layer 2.

**LSTM layer preference across cortex.** Earlier experiments found strong correspondence between layers of deep supervised networks and brain areas in visual (7; 5) and auditory (11) cortex. It is unclear whether a similar pattern should be expected for self-supervised networks such as the LSTM LM used here, where there is no clear hierarchy of layers. To examine this issue we visualized layer preference across cortex in Figure 5. Here a color is assigned to each significantly predicted voxel according to the relative encoding performance of each layer. Performance of the layer 1 model is shown using the red component of the color, layer 2 using green, and layer 3 using blue. Non-significantly predicted voxels appear gray, and voxels that are predicted equally well by each layer appear white.

Overall the performance of all layers is highly similar, rendering most voxels nearly white. Still, layer 2 provides the best predictions by a small margin, giving many voxels a green tint. However, auditory cortex (AC) shows a clear preference for layers 1 and 3, giving it a purple tint. Thus it seems that low-level speech processing (AC) is better modeled by layers 1 and 3, while high-level processing is better modeled by layer 2. This suggests that the middle layer of the LSTM LM is learning the highest-level representations, while layers 1 and 3, which are "closer" to the word embeddings, are learning lower-level representations. This finding is in stark contrast to experiments that used supervised networks (7; 5; 11). Those experiments found that high-level cortical areas were best modeled by higher layers of the network. Our results suggest that the highest-level representations in self-supervised models might emerge at the layer which is farthest from the input.

## 6    Conclusions

In this work, we effectively incorporate context representations for language using an LSTM LM. We observe that representations outperform state-of-the-art embedding based models and also show distinct behavior across different context lengths and LSTM layers. Our findings suggest that these models do indeed incorporate context and temporal order, albeit differently across layers. Finally, we show how our models explain the differences in language processing across different cortical regions, from low-level to high-level language areas.

As future work, we would like to explore and understand the information captured in the contextual representations that helps to model language processing in the cortex. Additionally, it would be worth exploring the roles played by different LSTM gates in developing these representations while incorporating context.

### Acknowledgments

We thank Jack Gallant, Wendy de Heer, Frederic Theunissen, and Thomas Griffiths for helping design the fMRI experiment and collecting the data used here; Brittany Griffin and Anwar Nuñez for segmenting and flattening cortical surfaces; and Niko Kriegeskorte for useful discussions. Data collection was supported by NSF grant IIS-1208203 and NIH NEI grant EY019684-01A1. This work was supported by grants from the Burroughs-Wellcome Fund and NVIDIA. We also acknowledge the Texas Advanced Computing Center (TACC) at The University of Texas at Austin for providing HPC resources that have contributed to the research results reported within this paper.

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
