[Supplementary Material]

# Supplement for Incorporating Context into Language Encoding Models for fMRI

Shailee Jain[1]    Alexander G Huth[1,2]

Departments of [1]Computer Science & [2]Neuroscience

The University of Texas at Austin

Austin, TX 78751

{shailee, huth}@cs.utexas.edu

October 2018

The supplementary shows additional subject flatmaps for the experiments in the main paper. For each subject, the top panel corresponds to Figure 3 in the main text and depicts the relative performance between the contextual and embedding-based encoding models. The middle panel corresponds to Figure 4 in the main text and shows the timescale selectivity across the cortex. Finally, the last panel corresponds to Figure 5 and depicts the LSTM layer preference across the cortex. The observed patterns are consistent across subjects, proving that the contextual representations extracted from an LSTM LM are effective for language encoding. Additionally, the contextualized encoding model can capture known timescale variance across the cortex.

# Subject S2

# Subject S3

# Subject S4