[Reviews · NeurIPS 2018]

Reviewer 1



This paper compares the embedding of a 3-layer LSTM to the neural responses of people listening to podcasts recorded via fMRI. The experiments vary the number of layers in the LSTM, and then context available to the LSTM and compare it to a context-free word embedding model. This is a strong paper, well written and clear. The results are thorough and there are a few interesting surprises. I have a few questions of clarification. 1) How do the authors account for the differences in number of words per TR due to differing word length and prosody? No mention of this adjustment is made. Other questions that would have been worth answering in this paper are the effect of corpora or LSTM variant on the performance. In particular, I see the argument for using the Reddit corpora, but wonder what the effect would have been to use another corpora of similar size. Here’s a great resources for finding such corpora, which includes some examples of transcribed speech (possibly a better fit to the podcasts being used as stimuli) http://courses.washington.edu/englhtml/engl560/corplingresources.htm I think many (most) NIPS attendees will not be familiar with the flat map, so more waypoints in the figures would be helpful (I see zero labels in F3, and only 3 in F4/5), as well as a link to a resource for interpreting flat maps. Also, BA doesn’t appear to be defined and it’s not clear which ROI it is assigned to in the flat maps, particularly in the LH. To my knowledge, the closest work to this is the 2014 Wehbe paper. It would be interesting to see a more thorough comparison to the results of that paper. There is only one sentence really comparing to previous results (line 258-9) and a deeper comparison would be nice to see. Minor comments: -Line 106: another critical flaw in some/most of these models is that the words are presented in isolation -Should cite the new Hale paper (https://arxiv.org/pdf/1806.04127.pdf Hale paper was *not* out when the current paper was submitted) -Line 173-176 are pretty unclear. Are you just describing the typical SEM? Or is it simply the standard deviation? If it’s a standard formula, this description can probably be dropped as it’s convoluted and not particularly helpful. Otherwise, stating it as an actual formula would be better.

Reviewer 2



I view this work as a very interesting extension of two recent exciting papers: Huth et al. (2016) and Baldassano et al. (2017). Essentially, the authors have developed a theory-based method for mapping out how context-dependent semantic information is distributed across the cortex. This is a big methods advance: whereas prior work aimed at constructing neural semantic maps has (for the most part) considered each moment's semantic content in isolation, this work provides a means of incorporating prior semantic content at different timescales. The authors clearly show that different brain areas "care" about semantic information at different timescales, and I like the overall LSTM-based approach. What dampens my enthusiasm slightly is that the ultimate conclusions seem to confirm (rather than specifically go beyond) similar to prior work from Uri Hasson's group, including the above-mentioned Baldassano et al. paper. I think there may be some way to interpret this set of results in a way that provides some key theoretical advance over that prior related work, and that's something I'd recommend that the authors focus on in their revision. Even if the theoretical impact does not quite live up to its potential, however, I still see this paper as providing an important methodological tool for creating context-dependent semantic brain maps from naturalistic experimental data. ** Update after reading author response and other reviews I stand by my assessment that this paper should be accepted. The authors have done a nice job addressing my (and the other reviewers') concerns.

Reviewer 3



The authors attempt to study how variable context length relates to the fMRI recordings of subjects who are listening to a story. They investigate variable context lengths through a 3-layer LSTM language model and test how well they can predict brain activity using representations from different layers and different context lengths. Quality: I completely agree that context is important in representing language in the brain. However, I believe this paper has not done enough to address very likely confounders. First off, it’s not clear how the encoding model is trained on the actual data set. As far as I understand, Figure 1 shows the pipeline at test time, not train time. How are the beta weights used for the fMRI data prediction in Section 5.1 obtained? If they are trained using the fMRI recordings and the corresponding stimuli from the data described in Section 2, then how did you handle multiple repetitions of a word in different contexts? I believe this is extremely important and can explain some of the results. The corruption of context results in Section 5.2 are expected because the LSTM is trained to model language, not random “nonsensical” sequences of text, so of course its representations will deteriorate. When the representations are bad, the prediction of fMRI data is also bad, so the encoding model performance decreases. A much more convincing control would be to look at the same word in two different contexts m1 and m2 (of same length), and compare the performance of the encoding model of predicting the brain activity of the word in one context from the LSTM representations of both m1 and m2. Then you can also vary the length of m1 and m2. The map results are speculative and are only for 1 subject -- why not show a mean over all subjects, or at least include all of the subject maps in the supplementary if these results are consistent? Clarity: It is not clear how the encoding model was trained, which is extremely important. In general, it would be good to see a lot more detail about the experiments than is offered. Also it is unclear what kind of correlation is used for the encoding model. Novelty: There does not seem to be methodological novelty. On the neuroscience side, using language models to show that context is important in the brain is also not a novel idea -- the paper even cites (though in an off-hand manner on line 41) one paper that uses an RNN LM to show that one can more accurately predict the brain activity for the next word by using the previous context (hidden layer of the RNN) than by using the embedding of the current word. The idea to study variable length context is interesting and novel, but I don’t believe that the experiments appropriately address this question. Significance: If my concerns are not correct and the brain map results are consistent across subjects, then this work is significant for the neuroscience community. Post rebuttal: The authors addressed two of my main concerns in their rebuttal: they showed results across additional subjects and presented results from a context swapping control experiment. However, in reading through the rebuttal and the paper an additional time, I have realized that there may be an additional confounder in the results, that is not controlled for in any of the experiments and could explain the main result. It is known that the LSTM word representations contain contextual information, and that these representations are sensitive to the nearby context length and word-order within this nearby context (https://arxiv.org/pdf/1805.04623.pdf). So the question is whether the improvement in encoding model performance with longer context is due to the increase in this contextual information in the LSTM representations (as the authors hypothesize), or to something else (a confounder)? One possible confounder is that increasing the context length of word W makes it more likely that word W will appear in this context, thereby allowing the LSTM representation of W in the longer context to be more influenced by W than it was in the shorter context. In this case, an encoding model trained to predict brain activity elicited by W from the longer context LSTM representations does not necessarily perform better due to the additional context, but possibly due to the fact that the LSTM representation is more similar to W. The more frequent W is, the worse the effect of this confounder could be. Considering that the presented results are over all words in the test set, including very frequent stop words, this alternative explanation could account for much of the improvement of the encoding model when the context length is increased. Control experiment: The randomized context experiment they perform in the original paper does not control for the number of repetitions of word W, and it also has another downside that I pointed out in my original review. The swapped context experiment, as performed by the authors, also does not control for this confounder, because the swapped context is chosen so that it “has the fewest words in common with the original text” [line 30 in rebuttal]. An experiment that would account for the presented confounder is similar to the swapped context experiment, but would have the same number of occurrences of word W in the original and the swapped context. Quantification of result significance: it is good that the authors present cortical map results for 3 additional subjects, however they are difficult to compare visually for consistency with the claimed results. In addition, it is difficult to judge how significant an increase in “encoding model performance” is with increased context without any notion of chance or hypothesis testing. For the above reasons, my score remains the same and I recommend the paper for rejection. I strongly disagree with the other two reviewers.